# Gallium nitride catalyzed the direct hydrogenation of carbon dioxide to dimethyl ether as primary product

Chang Liu[1], Jincan Kang[2], Zheng-Qing Huang [3], Yong-Hong Song[1], Yong-Shan Xiao[1], Jian Song[1], Jia-Xin He[1], Chun-Ran Chang [3✉], Han-Qing Ge[1], Ye Wang [2✉], Zhao-Tie Liu[1] & Zhong-Wen Liu[1✉]

The selective hydrogenation of $CO_2$ to value-added chemicals is attractive but still challenged by the high-performance catalyst. In this work, we report that gallium nitride (GaN) catalyzes the direct hydrogenation of $CO_2$ to dimethyl ether (DME) with a CO-free selectivity of about 80%. The activity of GaN for the hydrogenation of $CO_2$ is much higher than that for the hydrogenation of CO although the product distribution is very similar. The steady-state and transient experimental results, spectroscopic studies, and density functional theory calculations rigorously reveal that DME is produced as the primary product via the methyl and formate intermediates, which are formed over different planes of GaN with similar activation energies. This essentially differs from the traditional DME synthesis via the methanol intermediate over a hybrid catalyst. The present work offers a different catalyst capable of the direct hydrogenation of $CO_2$ to DME and thus enriches the chemistry for $CO_2$ transformations.

[1] Key Laboratory of Syngas Conversion of Shaanxi Province, School of Chemistry and Chemical Engineering, Shaanxi Normal University, Xi'an, China. [2] State Key Laboratory of Physical Chemistry of Solid Surfaces, Collaborative Innovation Center of Chemistry for Energy Materials, National Engineering Laboratory for Green Chemical Productions of Alcohols, Ethers and Esters, College of Chemistry and Chemical Engineering, Xiamen University, Xiamen, China. [3] Shaanxi Key Laboratory of Energy Chemical Process Intensification, School of Chemical Engineering and Technology, Xi'an Jiaotong University, Xi'an, China. ✉email: changcr@mail.xjtu.edu.cn; wangye@xmu.edu.cn; zwliu@snnu.edu.cn

The overdependence of the modern society on fossil fuels leads to huge $CO_2$ emissions, which induces adverse climate changes due to its greenhouse effects[1–4]. Provided the sustainable $H_2$ sources from renewable energies, such as wind or solar[5], a viable technology by hydrogenating $CO_2$ to hydrocarbons (HCs, e.g., $CH_4$, $C_2–C_4$ olefins, and gasoline) and oxygenates (methanol, ethanol, acetic acid, dimethyl ether (DME), etc.) can sustainably convert the renewable resources to chemicals and fuels. Thus, considerable efforts have been paid on this subject in recent years, and significant advances in the hydrogenation of $CO_2$ to CO, HCs, and oxygenates have been achieved[1,6,7]. Among these products, DME is a non-toxic, non-carcinogenic, and non-corrosive industrially important chemical used for a propellant of cosmetic products and the promising ultra clean fuel alternative to liquefied petroleum gas and diesel[8]. More importantly, the DME synthesis from $CO_2$ hydrogenation shows the highest efficiency, i.e., 97% of energy is stored in DME during its synthesis, which is higher than that stored in HCs or higher alcohols[9]. However, two-step reactions via the methanol intermediate are exclusively reported irrespective of direct or indirect processes[5,6,10,11]. Typically, metal/solid acid hybrid catalysts are the most efficient for the $CO_2$ hydrogenation to DME via the one-step coupling process. In this case, the metal-based catalysts, such as $Cu/ZnO/Al_2O_3$ catalyze the hydrogenation of $CO_2$ to methanol while the acid sites, such as HZSM-5 dehydrate the intermediate methanol to form DME.

As an important member of Group III nitrides, the thermodynamically stable wurtzite-structure gallium nitride (GaN), a well-known wide bandgap semiconductor with a fundamental bandgap energy of 3.4 eV[12,13], is quantitatively investigated as a revolutionary material due to its unique electronic and optical properties[14,15]. In the case of catalytic applications, GaN is increasingly investigated as a photocatalyst because of its high chemical and thermal stability[11,12]. Recently, GaN is found to be active and selective for the non-oxidative aromatization of light alkanes, such as methane[16–18], indicative of its catalytic ability for the activation of C–H bonds. Moreover, GaN shows acid property according to the density functional theory (DFT) calculations[19]. Therefore, considering these properties and the mechanistic understandings on the conversion of $CO_2$ to DME, GaN is expected to be a different type of catalyst for the direct hydrogenation of $CO_2$ to DME.

Herein, we demonstrate that the bulk GaN is an efficient catalyst for the selectively direct hydrogenation of $CO_2$ to DME, and DME was rigorously revealed as the primary product. Importantly, this differs from the traditional DME synthesis via the one-step coupling process over a hybrid catalyst, and a reasonable mechanism via the methyl and formate intermediates was proposed together with the DFT results. Moreover, the crystallite sizes of GaN, the addition of alkaline promoters, and the operating conditions had significant impacts on the catalytic performance. Under the optimal conditions, the space-time yield (STY) of DME as high as 2.9 mmol $g^{-1}_{GaN}$ $h^{-1}$ was obtained, and no deactivation was observed after a time-on-stream (TOS) of over 100 h.

## Results and discussion

**Catalytic performance.** To validate the possibility, we firstly investigated the commercial GaN powders (Alfa Aesar) as a catalyst for the hydrogenation of $CO_2$ in a fixed-bed reactor under the conditions of 300–450 °C, 2.0 MPa, $H_2/CO_2$ molar ratio of 3, gas hourly space velocity (GHSV) of 3000 mL $g^{-1}$ $h^{-1}$, and TOS of 40 h. The results indicate that the commercial GaN is really active for the hydrogenation of $CO_2$ and the $CO_2$ conversion continuously increases from about 5 to 35% as the reaction temperature increases from 300 to 450 °C (Fig. 1a). Moreover, the CO-free selectivity of DME is as high as ca. 80% at 300–360 °C, and CO and methanol are main by-products (Fig. 1). These results reveal that GaN can catalyze both the reverse water-gas shift (RWGS) reaction and the hydrogenation of $CO_2$ to oxygenates and HCs, the extent of which is clearly dependent on the reaction conditions. At a higher reaction temperature of 450 °C, the selectivity of DME sharply decreases to below 5% accompanied with the clearly increased selectivity of CO and $CH_4$. As a result, the highest DME STY of 0.56 mmol $g^{-1}_{GaN}$ $h^{-1}$ is achieved at 360 °C. Thus, GaN is a selective catalyst for synthesizing DME from the hydrogenation of $CO_2$.

To understand the active nature, multiple characterizations of the commercial GaN were performed. The results of X-ray diffraction (XRD) and transmission electron microscopy (TEM) reveal the pure wurtzite-structure GaN with a crystal size of 26.6 nm along the (110) direction (Fig. 2). The selected area electron diffraction pattern further indicates its polycrystalline structure exposed with different crystalline planes including (100) and (110) (Fig. 2c). Moreover, the surface Ga species are assigned to the GaN phase according to X-ray photoelectron spectroscopy (XPS), and the binding energies for Ga $2p$ are unchanged over the spent catalysts (Supplementary Fig. 1 and Supplementary Table 1). Thus, the wurtzite-structure GaN is tentatively speculated as the active phase for the selective hydrogenation of $CO_2$ to DME.

**Size effect.** Considering the commonly observed size effect of active phases in the heterogeneous catalysis, we synthesized bulk GaN with different crystal sizes by calcining the mixture of gallium nitrate and melamine at 800 °C for 1–4 h. The

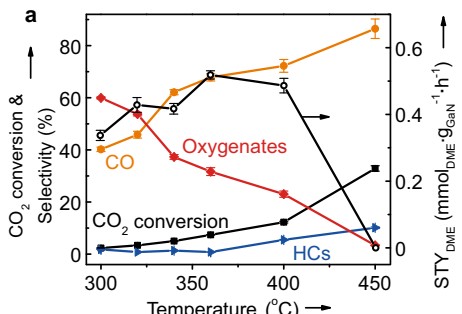

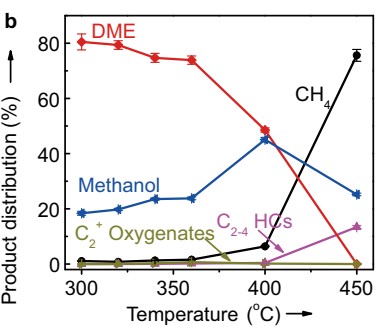

**Fig. 1 Catalytic performance of the commercial GaN for the hydrogenation of $CO_2$ at different temperatures. a** The $CO_2$ conversion, the selectivity of different products and the space-time yield of DME ($STY_{DME}$). **b** Detailed CO-free distribution of hydrocarbons (HCs) and oxygenates. Reaction conditions: $P = 2.0$ MPa, $H_2/CO_2 = 3$, gas hourly space velocity = 3000 mL $g^{-1}$ $h^{-1}$, and time on stream = 40 h. The error bar representing the relative deviation is within 5%.

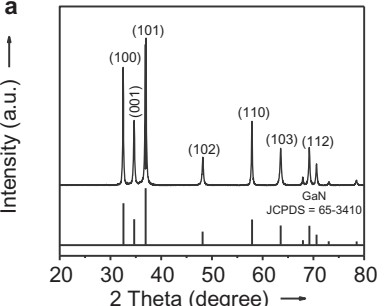
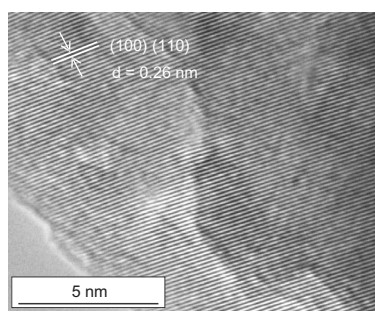
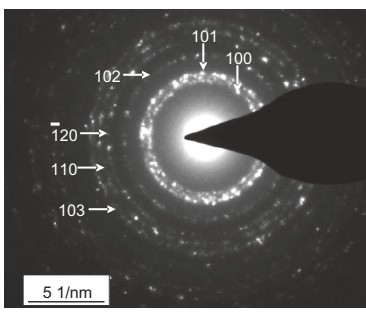

**Fig. 2 Structural characterizations of the GaN-26.6 catalyst. a** X-ray diffraction pattern. **b** Transmission electron microscopy image. **c** Selected area electron diffraction pattern.

characterization results of XRD (Supplementary Fig. 2), X-ray absorption fine structure analyses (XAFS, Supplementary Fig. 3), XPS (Supplementary Fig. 1 and Supplementary Table 1), and TEM analyses (Supplementary Fig. 4) confirm the formation of the pure wurtzite-structure GaN. Based on the Scherrer's formula and the (110) diffraction, the crystal size of the synthesized GaN samples is determined to be 7.4, 10.5, and 16.7 nm, respectively (Supplementary Table 2). Hereafter, the bulk GaN samples irrespective of the sources are denoted as GaN-$s$, where $s$ is the crystal size.

The size effect of GaN on the catalytic performance was evaluated with feed $H_2/CO_2$ ratios of 2 and 3, respectively, and the results are given in Fig. 3a, b. The continuously increased $CO_2$ conversion with decreasing the crystal size of GaN is clearly observed irrespective of $H_2/CO_2$ ratios although the impact is more pronounced at the $H_2/CO_2$ ratio of 3 (Fig. 3a). When the products are concerned, the same changing pattern is observed for the CO selectivity, i.e., the CO formation is favored over GaN with a smaller size. In the cases of the hydrogenated products detected by a flame ionization detector (FID), DME is the main product and is favored over GaN with a larger crystal size (Fig. 3b). Irrespective of the reaction conditions, the selectivity of HCs and $C_2^+$ oxygenates is very low. Moreover, the decreased selectivity of DME is always accompanied with the increased selectivity of methanol (Fig. 3b). Thus, the increased activity with decreasing the GaN size is mainly contributed by the enhanced RWGS reaction. Consequently, the highest selectivity and STY of DME are obtained over GaN-26.6.

To shed some lights on the nature of the size effect, the textural coefficient (TC) of the seven main XRD diffractions of GaN crystallite (Fig. 2a and Supplementary Fig. 2) was calculated according to the references[20,21]. From the definition of TC[17,18], it quantitatively represents the extent of the preferred crystal orientation, and the TC value for any crystalline plane over the ideal polycrystalline sample is equal to 1. Moreover, the TC value greater than 1 indicates the preferred crystal orientation, and the crystal orientation is more favorable with a higher TC value. Results (Supplementary Fig. 5) indicate that the (001), (100), and (110) planes with TC values of higher than 1 are the preferred crystal orientations over all of GaN catalysts. In the cases of GaN-7.4, GaN-10.5, and GaN-16.7, (001) plane is the most preferred orientation. In contrast, (110) and (100) planes are equally preferred over the GaN-26.6 catalyst. With increasing the crystal size of GaN from 7.4 to 26.6 nm, the crystal orientation of the (110) plane is increasingly preferred while a reversed changing pattern is observed for the (001) plane. With these understandings, the TC values for the preferred crystal orientations of the (100), (110), and (001) planes over different GaN samples were correlated with the STY of DME and CO calculated from the data in Fig. 3c–h. Irrespective of the $H_2/CO_2$ ratios, the

STY of DME is continuously increased with increasing the TC value of the (110) plane while the STY of CO is almost linearly increased with increasing the TC value of the (001) plane. In the case of the (100) plane, there exists no simple relationship between the TC value and the STY of DME or CO. These results reveal that the crystal size effect of GaN catalysts on the selectivity of different products during the hydrogenation of $CO_2$ is essentially originated from the changing in the preferred crystal orientation of different planes. Thus, the higher STY of DME over the GaN catalyst with a larger crystal size can be explained as the more preferred crystal orientation of the (110) plane. Moreover, the higher STY of CO over the GaN catalyst with a smaller crystal size is due to the more preferred crystal orientation of the (001) plane.

**Catalytic stability and effect of basic promoter**. To probe the stability of GaN catalysts, GaN-26.6 was representatively evaluated under optimized conditions for a TOS of 100 h. The results (Fig. 4) show that there is no observable deactivation after the induction period of about 12 h. The origin of the induction period may be due to the loss of acidity (Supplementary Fig. 6a and Supplementary Table 3). Therefore, a long-term stability of GaN as a catalyst for the selective hydrogenation of $CO_2$ to DME is reasonably expected. To further enhance the yield of DME, $CaCO_3$ as a basic promoter was physically mixed with GaN-26.6 with different molar ratios. As shown in Supplementary Fig. 7, the highest $STY_{DME}$ of 2.9 mmol $g^{-1}_{GaN}$ $h^{-1}$ is obtained over the catalyst with a $CaCO_3$ to GaN-26.6 molar ratio of 1, which is clearly higher than that over the Cu-based hybrid catalysts under similar reaction conditions (Supplementary Table 4).

**DME as the primary product**. The selectivity of CO is clearly higher at a higher $H_2/CO_2$ ratio (Fig. 3a), indicating that the RWGS reaction is enhanced under the conditions with a higher partial pressure of $H_2$. This is different from the common observation, i.e., a lower selectivity of CO at a higher feed $H_2/CO_2$ ratio, over traditional oxide catalysts for the hydrogenation of $CO_2$ to methanol[22], gasoline fuel[23] or aromatics[24], which is explained as the secondary hydrogenation of CO. To find out the reason, the hydrogenation of CO was comparatively studied over GaN with different crystal sizes (Fig. 5). Under the same reaction conditions but a much lower GHSV of 1000 mL $g^{-1}$ $h^{-1}$, the conversion rates of CO are still significantly lower than those of the $CO_2$ hydrogenation although the product distributions are very similar. Thus, the secondary reaction, i.e., the hydrogenation of CO produced from the hydrogenation of $CO_2$, is negligible. These facts suggest that the GaN-catalyzed $CO_2$-to-DME and RWGS are competitively parallel reactions. Moreover, DME is possibly formed as the primary product from the direct

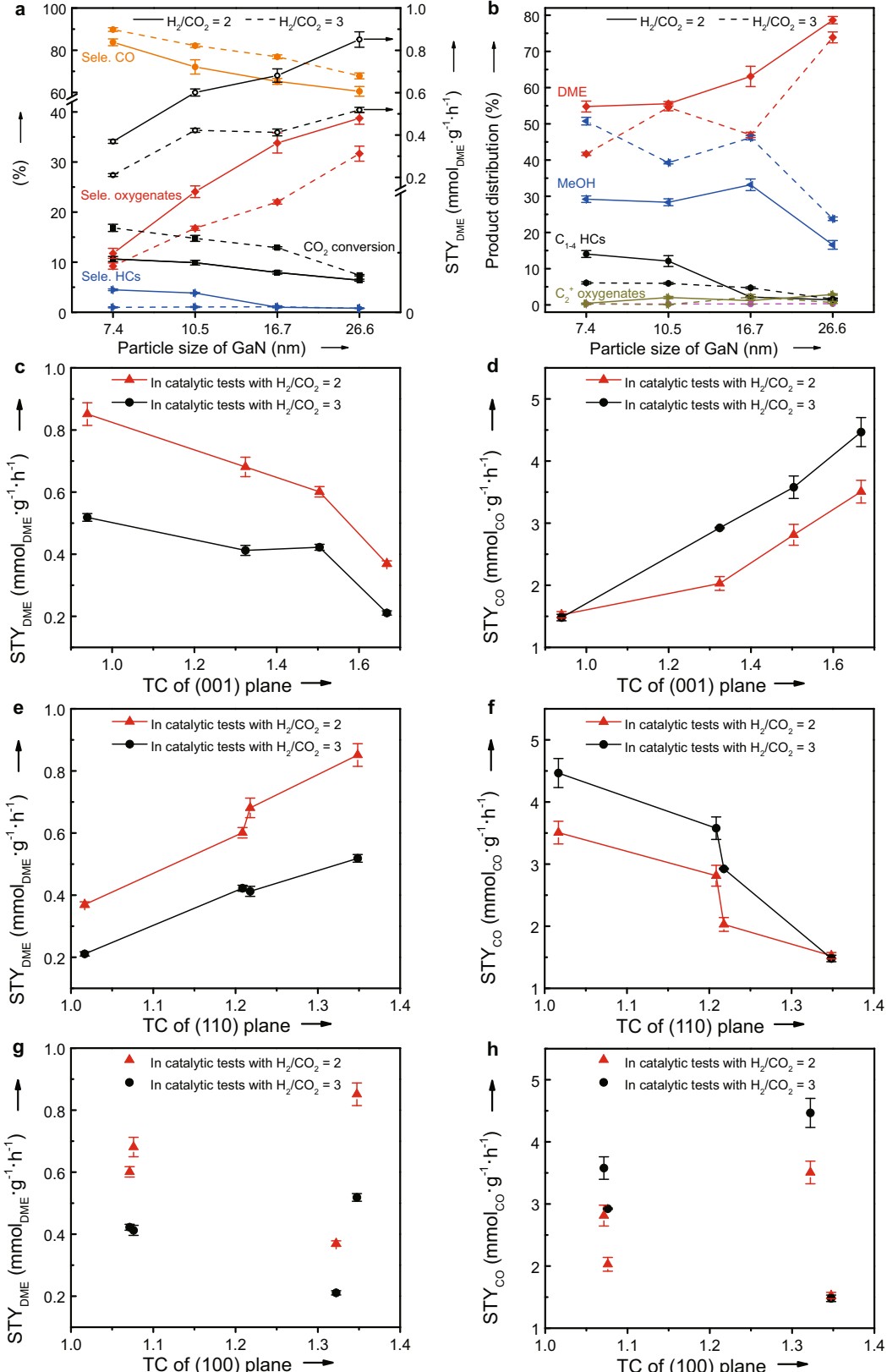

**Fig. 3 Effect of GaN crystal properties on the catalytic performance. a** Size effect on the $CO_2$ conversion, selectivity (Sele.) of different products and the space-time yield of DME ($STY_{DME}$). **b** The CO-free distribution of hydrocarbons (HCs) and oxygenates for the hydrogenation of $CO_2$; **c–h** The correlation between the textural coefficients (TC) of (001), (110), and (100) planes with $STY_{DME}$ and the space-time yield of CO ($STY_{CO}$). Reaction conditions: $P =$ 2.0 MPa, $T =$ 360 °C, $H_2/CO_2 =$ 2 or 3, gas hourly space velocity = 3000 mL g$^{-1}$ h$^{-1}$, and time on stream = 40 h. The error bar representing the relative deviation is within 5%.

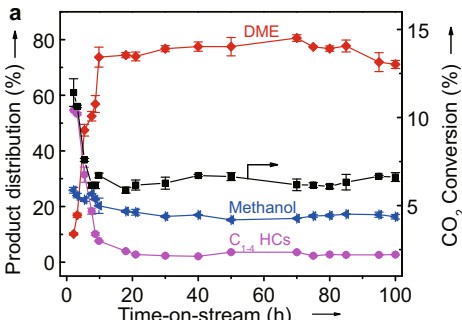
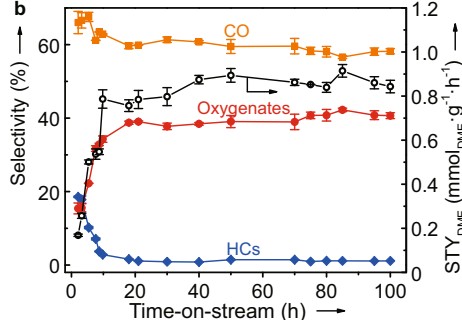

**Fig. 4 The representative long-term stability results for the hydrogenation of $CO_2$ over GaN-26.6. a** The $CO_2$ conversion and the CO-free distribution of the hydrocarbons (HCs) and oxygenates. **b** The selectivity of different products and space-time yield of DME ($STY_{DME}$). The error bar is for 5% relative deviation. Reaction conditions: $P = 2.0$ MPa, $T = 360$ °C, $H_2/CO_2 = 2$, and gas hourly space velocity = 3000 mL g$^{-1}$h$^{-1}$.

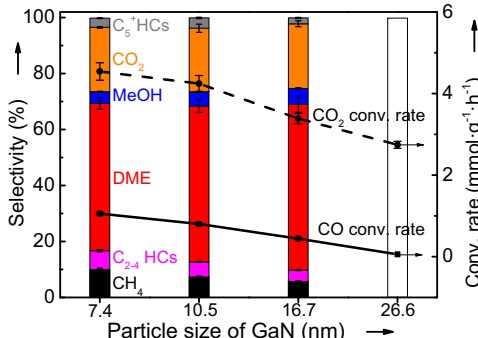

**Fig. 5 Catalytic performance for the $CO/CO_2$ hydrogenation over the GaN catalysts with different particle sizes.** Reaction conditions: $T = 360$ °C, $P = 2.0$ MPa, gas hourly space velocity = 1000 mL g$^{-1}$h$^{-1}$ for the CO hydrogenation and 3000 mL g$^{-1}$h$^{-1}$ for the $CO_2$ hydrogenation, $H_2/CO$ ($CO_2$) = 2, and time on steam = 40 h. HCs hydrocarbons, Conv conversion. The error bar indicating the relative deviation is within 5%.

hydrogenation of $CO_2$. To confirm this, the hydrogenation of $CO_2$ was studied by changing the contact time ($W/F$, $W$ for the catalyst weight and $F$ for the flow rate of reactants), and the results are given in Fig. 6a. With increasing the contact time from 1.2 to 2.4 s g mL$^{-1}$, the selectivity of DME is significantly decreased together with a pronounced decrease in the selectivity of CO and an obviously increased selectivity of methanol and HCs. If the contact time is further increased to 4.5 s g mL$^{-1}$, changes in the selectivity of any products are very limited. These results reveal that DME is very possibly the primary product while methanol and HCs are the secondary products. By co-feeding DME and $H_2O$ (Supplementary Fig. 8a), it is confirmed that GaN can catalyze the hydrolysis of DME to methanol ($CH_3OCH_3 + H_2O = 2CH_3OH$, Supplementary Fig. 8a). Moreover, the dehydration of methanol to DME occurs significantly by using methanol as the reactant (Supplementary Fig. 8b). If methanol and $H_2O$ are co-fed into the reactor, the decomposition of methanol and/or the steam reforming of methanol dominate to produce the reformate (Supplementary Fig. 8c), which are consistent with the high selectivity of CO or $CO_2$ at a higher reaction temperature in the cases given in Supplementary Fig. 8a, b. These results agree well with the reversible nature of the acid-catalyzed dehydration of methanol to DME[25,26], indicating the presence of acid sites over GaN. To directly reveal the acid property, GaN catalysts were characterized by the pyridine adsorbed infrared (IR) spectroscopy (Supplementary Fig. 9 and Supplementary Table 5) and $NH_3$ temperature-programmed desorption ($NH_3$-TPD, Supplementary Fig. 6b and Supplementary Table 3). By correlating the acidity of GaN with the $STY_{DME}/STY_{MeOH}$ ratio

determined from the data in Fig. 3, a higher amount of acid sites over the catalyst, especially Brønsted acid sites, is found to favor the formation of methanol rather than DME (Supplementary Fig. 10). This is essentially different from the observation for the hydrogenation of $CO_2$ over Cu-based hybrid catalysts, i.e., a higher amount of Brønsted acids improving the selectivity of DME, which is explained as that the dehydration of methanol as the secondary reaction is enhanced by a higher amount of Brønsted acids[27]. Thus, DME is concluded as the primary product over GaN catalysts for the hydrogenation of $CO_2$. On the contrary, methanol is formed as a secondary product from the hydrolysis of DME catalyzed by the acid sites over GaN.

To reveal the reaction profile of the $CO_2$ hydrogenation, the temperature-programmed surface reaction (TPSR) over GaN-26.6 was performed under the conditions of 2.0 MPa and $H_2/CO_2$ ratio of 2. From the results shown in Fig. 6b, the formation of CO is monotonically increased with increasing the temperature, indicating that the consumption of CO via any secondary reactions including its hydrogenation is insignificant. This is well agreeable with the low activity of GaN catalysts for the CO hydrogenation (Fig. 5). Although the temperature for the appearance of DME and methanol cannot be differentiated under the present conditions, the maxima of DME and methanol are observed from the TPSR profiles. More importantly, the formation of DME reaches the maximum at 359 °C while the temperature for the maximum of methanol is significantly higher (385 °C). This strongly supports that the increased formation of methanol is originated from the consumption of DME. Thus, both CO and DME are primary products for the GaN-catalyzed hydrogenation of $CO_2$. Moreover, CO is produced from the RWGS reaction while DME is the consequence of the direct hydrogenation of $CO_2$.

As shown in Fig. 6b, the profile for the formation of methane during TPSR is different from other products. It is continuously increased with increasing the temperature from about 250 to ~360 °C. When the temperature is further increased from ~360 °C, the formation of methane is first decreased and then is increased again, leading to the minimum at about 420 °C. This suggests that methane may be formed from different reaction routes depending on the reaction temperature. At the temperature of below 420 °C, methane may be mainly produced from the secondary reactions of oxygenates including DME and methanol catalyzed by acid sites, which is essentially the same to the reaction of DME/methanol to HCs. This is directly supported by the increased selectivity of $CH_4$ with increasing the contact time (Fig. 6a), which favors the secondary reactions at a longer contact time. Alternatively, the direct hydrogenation of $CO_2$ to methane may be predominant at a temperature of higher than 420 °C, which is proved from the very high selectivity of $CH_4$ for the hydrogenation of $CO_2$ at 450 °C (Fig. 1b).

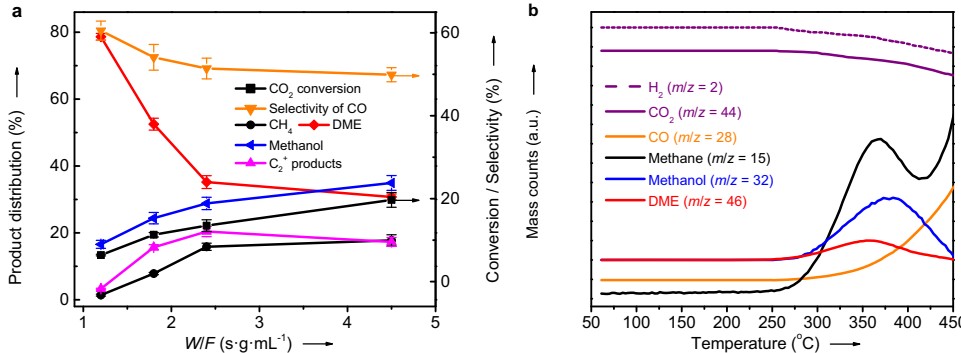

**Fig. 6 Experimental results confirming DME as the primary product. a** Effect of the contact time on the catalytic behavior of the hydrogenation of $CO_2$ over GaN-26.6 at 360 °C (The contact time is expressed as the catalyst weight ($W$) divided by the flow rate of the feed gases ($F$). The error bar representing the relative deviation is within 5%). **b** Temperature-programmed surface reaction profiles of the hydrogenation of $CO_2$ over GaN-26.6 under the conditions of $P = 2.0$ MPa and $H_2/CO_2 = 2$ ($m/z$: mass-to-charge ratio).

**Intermediate species over GaN surface**. The GaN-catalyzed hydrogenation of $CO_2$ produces DME as the primary product, which is completely different from those over the hybrid catalysts. To understand the mechanism, the intermediates for the hydrogenation of $CO_2$ over GaN must be determined. Thus, operando diffuse reflectance infrared Fourier transform spectroscopy (DRIFTS) of $CO_2$ hydrogenation was performed. DRIFTS results indicate that carboxylate, carbonate, and methyl species are detected at the initial stage of the reaction (Supplementary Fig. 11 and Supplementary Table 6). The bands assigned to bicarbonate and bi-dentate formate species are observed after a TOS of about 5 min. Moreover, the intensity of the IR band for the methyl group is significantly decreased while those of bicarbonate and formate species are still clearly observable with increasing the TOS. In the case of the IR band at 1456 cm$^{-1}$ appearing after about 5 min, its intensity is slightly increased with increasing TOS (Supplementary Fig. 12), which can be assigned to the characteristic C–H bond vibrations of the absorbed DME. According to the references[28–30], $CO_3^{2-}$ and $HCO_3^-$ species are likely intermediates for the formation of CO via the RWGS reaction. Moreover, the HCOO$^*$ species are important intermediates for the formation of oxygenates during the $CO_2$ hydrogenation[31,32]. By analyzing the time-evolved absorbance of the typical IR bands (Supplementary Fig. 12), the HCOO$^*$ species may be originated from the hydrogenation of the COO$^*$ or $CO_3^{2-}$ species. However, a previous study indicates that the hydrogenation of $CO_3^{2-}$ to CO and $H_2O$ via $HCO_3^-$ is more favorable than the catalytic conversion of $CO_3^{2-}$ to HCOO$^*$ at the reaction temperature of higher than 300 °C[33]. Therefore, the HCOO$^*$ species is more likely formed from the hydrogenation of the COO$^*$ species, which is generally consistent with the results over ZnO-ZrO$_2$[32], Ru/ CeO$_2$[33], and Cu/CeO$_2$/TiO$_2$[34].

**DFT calculations**. To validate the reaction mechanism of GaN-catalyzed $CO_2$-to-DME, the key steps of $CO_2$ hydrogenation were investigated by DFT calculations. As revealed by the calculated surface energies (Supplementary Fig. 13), GaN(100) and GaN (110) are more stable than GaN(001). Moreover, GaN(001) favors the formation of CO rather than DME during the hydrogenation of $CO_2$ (Fig. 3c, d). Therefore, the GaN (100) and (110) surfaces are considered in the following calculations. As shown in Supplementary Figs. 14 and 15 and Supplementary Tables 7 and 8, H2 molecules dissociate at Ga–N pairs in a heterolytic pathway with low activation barriers of 0.09 eV on GaN(100) and 0.17 eV on GaN(110). In the case of $CO_2$, it binds strongly on both surfaces (−1.72 eV on GaN(100) and −1.61 eV on Ga(110),

Supplementary Fig. 16) by forming the bent COO$^*$ species ($^*$ represents an adsorbed state), which is also experimentally proved by DRIFTS results (Supplementary Table 6). Noteworthy, the slight difference between the adsorption energies of H$_2$ and $CO_2$ (<0.40 eV) indicates that the adsorption of the two reactants on GaN surfaces is comparable, which paves the way for the facile hydrogenation of $CO_2$ on the catalyst surface. In contrast, CO is weakly adsorbed on GaN surfaces (−0.61 eV on GaN(100) and −0.65 eV on Ga(110), Supplementary Fig. 16) in comparison with the H$_2$ dissociative adsorption, leading to a low coverage of CO on GaN surfaces, which is consistent with the lower activity for the CO hydrogenation (Fig. 5).

To differentiate the intermediates, the hydrogenation of $CO_2$ to carboxyl (COOH$^*$) and formate (HCOO$^*$) was studied on GaN (100) and (110) surfaces (Supplementary Fig. 17 and Supplementary Tables 7 and 8). On the GaN(100) surface, the calculated reaction energies ($\Delta E$) and activation energies ($E_a$) manifest that the hydrogenation of $CO_2$ to COOH$^*$ is more favorable than that to HCOO$^*$, in which $E_a$ of the former is 0.25 eV lower than that of the latter. In the case of GaN(110) surface, the formation of HCOO$^*$ is more advantageous with a lower $E_a$ of 0.87 eV than that for the formation of COOH$^*$ (1.87 eV). Thus, the pathway for forming COOH$^*$ on GaN(100) and HCOO$^*$ on GaN(110) is more favorable, indicating the different tendencies for the first-step hydrogenation on GaN (100) and (110) surfaces.

A detailed pathway for the $CO_2$ hydrogenation to generate CH$_3^*$ on GaN(100) was studied by calculating standard Gibbs free energies at 360 °C (Fig. 7a). The adsorbed COOH* firstly dissociates into CO$^*$ and OH$^*$ with a moderate Gibbs free energy of activation ($G_a$, 1.15 eV). Subsequently, CO$^*$ is hydrogenated to CHO$^*$, CH$_2$O$^*$, and CH$_2$OH$^*$ with reasonable $G_a$ values. The succeeding reaction of CH$_2$OH$^*$ can be either hydrogenated to CH$_3$OH$^*$ or be dissociated into to CH$_2^*$ and OH$^*$. However, $G_a$ for the former (1.28 eV) is greatly higher than that of the latter (0.26 eV), indicating that the dissociation of CH$_2$OH$^*$ into CH$_2^*$ and OH$^*$ is significantly more favorable than the formation of methanol. This well explains the experimental finding that methanol is not the primary product of the $CO_2$ hydrogenation. The dissociated CH$_2^*$ can be further hydrogenated to CH$_3^*$ with a moderate $G_a$ of 1.16 eV, which is consistent with the experimentally detected methyl by DRIFTS (Supplementary Fig. 11). On GaN(110), the detailed Gibbs free energy profiles of the $CO_2$ hydrogenation to HCOO$^*$ were also calculated (Fig. 7b). At the temperature of 360 °C, $G_a$ for $CO_2$ to HCOO$^*$ is only 0.77 eV. The transformation of mono-dentate formate (*mono*-HCOO$^*$) to bi-dentate formate (*bi*-HCOO$^*$) is thermodynamically favorable

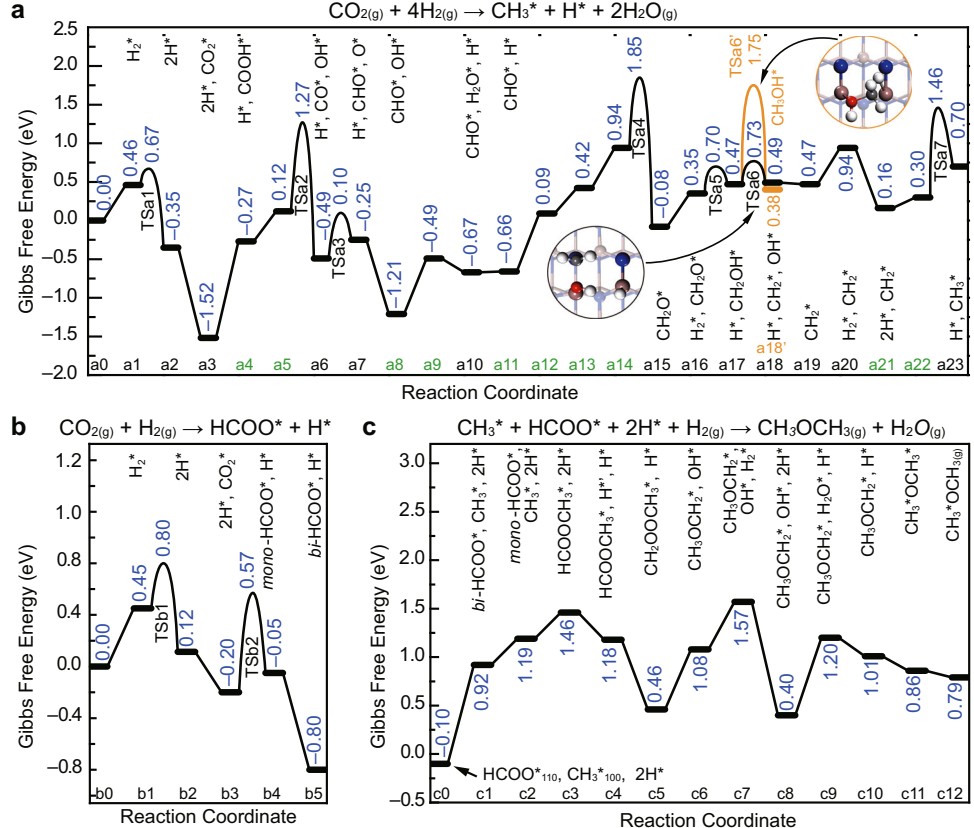

**Fig. 7 The DFT calculation results. a** Gibbs free energy diagram of the $CO_2$ hydrogenation to methyl ($CH_3^*$) on the GaN(100) surface. **b** Gibbs free energy diagram of the $CO_2$ hydrogenation to formate ($HCOO^*$) on the GaN(110) surface. **c** Gibbs free energy diagram for the coupling of $HCOO^*$ and $CH_3^*$ to DME on the (110)/(100) interface. The corresponding structures of initial states (IS), transition states (TS), and final states (FS) are displayed in Supplementary Figs. 18–21. The zero-energy reference corresponds to the sum of Gibbs free energies (at 360 °C) of $H_{2(g)}$, $CO_{2(g)}$, and the respective clean surface. The state notations colored in green reflect that the reactions between these neighboring states are the diffusion of the surface adsorbents.

with a Gibbs free energy ($\Delta G$) of $-0.75$ eV, which is experimentally supported by DRIFTS results over GaN catalysts (Supplementary Fig. 11).

To produce DME, the $CH_3^*$ and $HCOO^*$ species may combine to form $HCOOCH_3^*$ at the (100)/(110) interface, followed by the stepwise hydrogenation and dehydration to give the final product DME (Fig. 7c). From the overall Gibbs free energy profile, it is smooth with the largest $\Delta G$ of 0.80 eV (from c8 to c9), indicating that the proposed mechanism is plausible at the temperature of 360 °C. Noteworthy, the whole reaction pathway does not involve the $CH_3O^*$ species, which further confirms that methanol is not the primary product for the hydrogenation of $CO_2$ to DME. Thus, DFT calculations provide a possible pathway for the formation of DME and well explain why DME rather than methanol is formed as the primary product on GaN surfaces, i.e., the difficult hydrogenation of $CH_2OH^*$ to $CH_3OH^*$ and the unfavorable formation of $CH_3O^*$.

**Reaction mechanism**. Based on the experimental and DFT calculation results, the possible mechanism is proposed in Fig. 8. For starting the reaction, $CO_2$ molecules are activated on GaN as the bent $COO^*$ species while $H_2$ is adsorbed dissociatively on Ga–N Lewis pairs. Depending on the different surfaces of GaN, the ensuing hydrogenation of the $COO^*$ species occurs simultaneously in two routes. One path (the orange arrow in Fig. 8) is the hydrogenation of $COO^*$ on the GaN (110) surface to produce $HCOO^*$. The other path (the blue arrow in Fig. 8) is the hydrogenation of $COO^*$ to $CH_3^*$ on the GaN (100) surface via the intermediates of $^*CHO$, $^*CH_2OH$, and $CH_2^*$, which is

supported by DRIFTS, DFT calculations, and reference results[35]. Finally, $HCOO^*$ and $CH_3^*$ are coupled at the (100)/(110) interface to form DME via a series of hydrogenation and dehydration steps (the black arrow in Fig. 8).

In summary, we demonstrated that wurtzite-structure GaN is active, stable, and selective for the direct hydrogenation of $CO_2$ to DME. The larger GaN nanocrystals or the addition of $CaCO_3$ as a promoter can clearly enhance the catalytic performance. Significantly, the activity of GaN for the hydrogenation of CO is much lower than that for the hydrogenation of $CO_2$ although the product distributions are very similar. The transient, operando DRIFTS, and DFT results rigorously reveal that DME is formed as the primary product via the coupling of $CH_3^*$ and $HCOO^*$. Moreover, HCs and oxygenates including methanol are produced via the secondary reactions catalyzed by the acid sites over GaN. This clearly differs from the traditional one-step coupling process via the methanol intermediate over a typical hybrid catalyst. These findings may open up a different catalytic route for the directly efficient utilization of $CO_2$.

## Methods
**Preparation of catalysts**. The GaN catalysts were synthesized by calcining the powdery mixture of gallium nitrate together with melamine at 800 °C in a flow of nitrogen. Firstly, gallium nitrate ($Ga(NO_3)_3$, Macklin) and melamine ($C_3H_6N_6$, Kermel) at a Ga/N molar ratio of 1 were mixed by the mortar-mixing method. Subsequently, the mixed powers were calcined in a tubular oven at 800 °C with a temperature ramp of 1 °C/min in a nitrogen flow of 100 mL/min. The duration of the calcination was 1, 2, and 4 h, respectively. After the calcination, the sample was further calcined in air at 550 °C for 2 h. The obtained sample was denoted as GaN-s, where s was the particle size of the GaN crystallite measured by XRD.

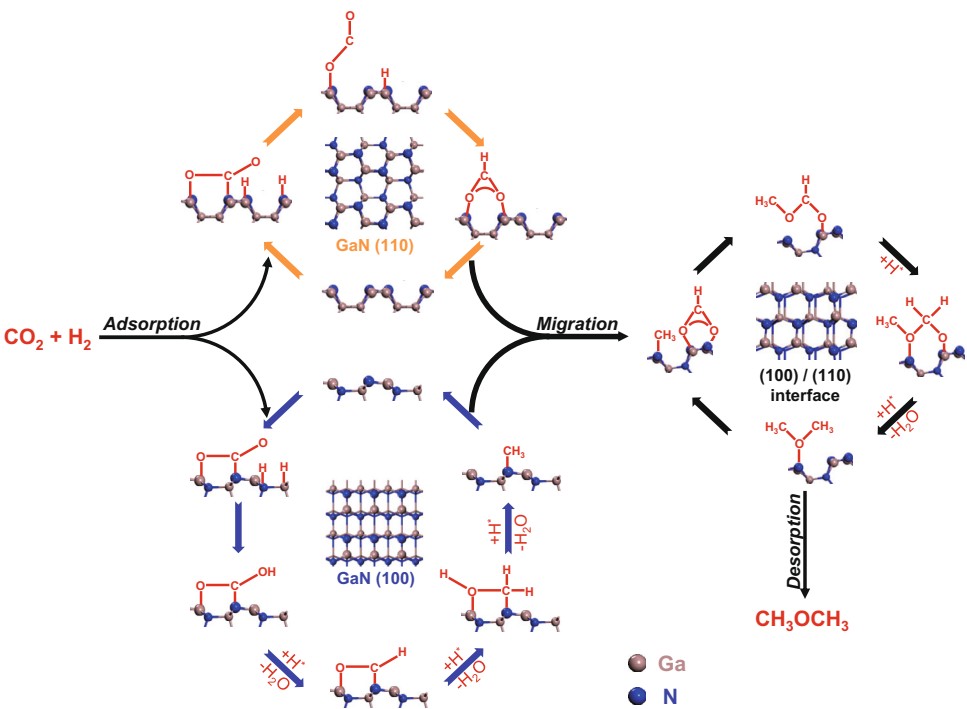

**Fig. 8 The mechanism for CO₂ to DME catalyzed by GaN.** The route with orange arrows: elementary steps for the formation of HCOO* over the (110) plane. The route with blue arrows: elementary steps for the formation of CH₃* over the (100) plane. The route with black arrows: elementary steps for the formation of DME over the (100)/(110) interface.

The promoted GaN catalysts were also prepared by the mortar-mixing method. The promoters employed in this work were $K_2CO_3$ (Aladdin), $MgCO_3$ (Aladdin), $CaCO_3$ (Macklin), CaO (Macklin), Ca(OH)₂ (Aladdin), and CaAc₂ (Aladdin), respectively. After the mortar-mixing, the mixture of GaN and the promoter was calcined in air at 400 °C for 2 h. The obtained catalysts were denoted as $y$A/GaN, where $y$ is the molar ratio of the promoter to GaN and A is the promoter.

**Characterization techniques**. XRD patterns were obtained on a Bruker D8 Advance X-ray diffractometer with a monochromatic Cu/Kα radiation (40 kV, 40 mA). The samples were scanned from 5 to 80° (2θ) with a step size of 0.02° and a counting time of 0.2 s per step. To determine the average crystallite sizes, we used the half-width of (110) peak in the diffraction pattern and the Scherrer's equation:

$$L = (C\lambda)/(\beta \cos\theta) \quad (1)$$

where C is a constant (0.89), λ is the wavelength of the X-ray (0.154 nm), β is the full-width at half-maximum of a peak in the diffraction pattern, θ is the Bragg angle, and L is the volume-averaged size of the crystallites. The β and θ were measured using JADE software.

To determine the preferred crystallographic orientations, TC is calculated for different crystallite planes by the Harris formula [17,18]:

$$TC(hkl) = \frac{I(hkl)/I_0(hkl)}{\frac{1}{N}\sum_{j=1}^{N} I(h_jk_jl_j)/I_0(h_jk_jl_j)} \quad (2)$$

where (hkl) is the specific plane, $I_0(hkl)$ is the intensity of the (hkl) plane in the standard wurtzite-structure GaN crystallite, and I(hkl) is the intensity of the (hkl) plane based on the XRD patterns of the sample. The total number of planes, N, is determined to be 7 from the prominent and well separated XRD peaks.

XAFS analyses of the Ga κ-edge was measured at the BL12B-a beamline of the National Synchrotron Radiation Laboratory in the total electron yield mode under a vacuum better than $5 \times 10^{-6}$ Pa. The beam from the bending magnet was monochromatized utilizing a varied line-spacing plane grating and refocused by a toroidal mirror.

XPS analyses were performed with an Axis Ultra spectrometer (Kratos Analytical Ltd.) using an Ag monochromatic X-ray source (Ag Kα = 1486.6 eV) at room temperature in a high vacuum environment (~$5 \times 10^{-9}$ torr). All the binding energies were calibrated to the containment carbon C 1 s peak (284.8 eV).

TEM observations were performed on a JEM 2100 electron microscope (JEOL, Japan) operated at 200 kV. The powdered sample was ultrasonically dispersed in ethanol and deposited on a copper grid prior to the measurements.

Pyridine adsorbed IR measurements were performed on a Nicolet iS50 instrument equipped with a deuterated triglycine sulfate detector. The GaN samples were mixed with KBr with a mass ratio of GaN/KBr = 1/1 and 20 mg of

the mixed powders were pressed into a self-supported wafer and placed in an in situ IR cell. The absorbance spectra were measured by collecting 32 scans with resolution of 4 cm⁻¹. After pretreatment under vacuum at 400 °C for 3 h, the sample was cooled to 50 °C. Spectra of degassed samples were collected as background. Pyridine was adsorbed at 50 °C for 0.5 h. Then, the IR cell was heated to 150 °C under vacuum for 0.5 h, and spectra of pyridine adsorbed samples were collected.

NH₃ temperature-programmed desorption (NH₃-TPD) was measured on a Micromeritics Autochem 2920 instrument. Around 100 mg of sample was put into a quartz reactor and was pretreated in Ar at 550 °C for 1 h. The absorption of NH₃ was realized at 50 °C in a 10 vol% NH₃/Ar flow of 20 mL/min for 1 h. Then the gas was shifted to Ar (30 mL/min) and the sample was purged for 2 h. Subsequently, the desorption of NH₃ was realized in Ar (30 mL/min) with a ramp of 5 °C/min until 900 °C. The desorbed NH₃ was measured by a mass spectrometer (Hiden QIC-20).

TPSR characterization was carried out on a Micromeritics Autochem 2950 instrument. Around 500 mg of sample was loaded in a stainless steel reactor. It was pretreated at 400 °C in a mixed gas of CO₂/H₂/Ar = 32/64/4 for 2 h and in Ar for 1 h, respectively. After cooling down to 50 °C in Ar, the reactants with a molar ratio of CO₂/H₂/Ar = 32/64/4 was introduced at a flow rate of 30 mL/min. By keeping the pressure at 2.0 MPa, the TPSR experiments were started from 50 to 450 °C with a ramp of 1 °C/min, and the mass signals of CO₂, H₂, Ar, CO, CH₄, DME, and CH₃OH were recorded by a mass spectrometer (Hiden QIC-20).

Operando DRIFTS measurements were applied to probe the reaction intermediates over GaN catalysts. The spectra were obtained by collecting 16 scans with a resolution of 4 cm⁻¹ on a Nicolet iS50 instrument equipped with a MCT detector. About 0.2 g of the GaN sample mixed with KBr with a mass ratio of GaN/KBr = 1/30 was placed in the in situ cell (Diffuse IR, PIKE company, American). Then, the sample was pretreated at 400 °C sequentially in the mixed gases of CO₂/H₂/Ar = 32/64/4 for 2 h and in Ar for 1 h. After cooling the catalyst down to 360 °C, the background spectrum was recorded in an Ar flow. The spectra for the CO₂ hydrogenation reaction were realized at 360 °C, 0.1 MPa, CO₂/H₂/Ar = 32/64/4, and flow rate = 20 mL/min.

**Reaction procedure and product analyses**. The catalytic tests of the CO₂ hydrogenation were conducted using a quartz-coated stainless steel fixed-bed reactor (id = 5.9 mm). The mixed gases with a H₂/CO₂ molar ratio of 2 or 3 was fed to the reactor, and 4 vol% of Ar in the mixed gases was used as an internal standard. The reaction was conducted at P = 2.0 MPa, T = 300–450 °C, and GHSV = 800–3000 mL g⁻¹ h⁻¹. To estimate experimental errors, catalytic tests of representative catalysts were repeated at least twice. The reactants and effluent products were analyzed on-line using a GC-9560 gas chromatograph (Huaai Company) equipped with a thermal conductivity detector (TCD) and FID. The effluents of Ar, CO, CH₄, and CO₂ were separated by a TDX-01 column and were analyzed by

TCD. The separation of DME, MeOH, and $C_{1-5}$ HCs were performed on a Plot-Q column (Bruker) and were measured by FID. The $C_2^+$ oxygenates (Oxys) i.e., ethanol, propanol, acetic acid, propionic acid, butyric acid, etc., and $C_{6-12}$ HCs were collected in a cold trap and were off-line analyzed on a GC-2010 gas chromatograph (Shimadzu) equipped with a SH-Rtx-Wax column and an FID. The $CO_2$ conversion, selectivity of CO, HCs, and Oxys, the distribution of products based on carbon numbers (CO-free products, i.e., methane, $C_{2-4}$ HCs, $C_5^+$ HCs, MeOH, DME, and $C_2^+$ Oxys), the STY of products, and the conversion rate of $CO_2$ was calculated with the following equations.

$$CO_2 \text{ conversion} = [F_{in}(CO_2) - F_{out}(CO_2)]/F_{in}(CO_2) \times 100\% \quad (3)$$

$$\text{Selectivity of CO, HCs, or Oxys} = F_{out}(CO/HC_s/Oxy_s)/[F_{in}(CO_2) - F_{out}(CO_2)] \times 100\% \quad (4)$$

$$\text{Distribution of product } A = F_{out}(A)/F_{out}(\text{Carbon}_{FID}) \times 100\% \quad (5)$$

$$STY_A = [F_{out}(A)/V_m]/m_{GaN} \quad (6)$$

$$\text{Conversion rate of } CO_2 = [F_{in}(CO_2) - F_{out}(CO_2)]/V_m/m_{GaN} \quad (7)$$

where $F_{in}$ is the flow rate in the inlet and $F_{out}$ is the flow rate in the outlet. $A$ is one of the products detected by FID. $F_{out}(\text{Carbon}_{FID})$ is the flow rate of total carbon atoms of the hydrogenated products detected by FID in the outlet. $STY_A$ is the STY of product $A$. $V_m$ is the molar volume of an ideal gas at the standard temperature and pressure, which 22.4 L/mol is used for calculations. $m_{GaN}$ is the weight of GaN in the catalyst bed.

The CO hydrogenation was also carried out in the same reactor by feeding CO/$H_2$/Ar with a molar ratio of 32/64/4 under the conditions of $P = 2.0$ MPa, $T = 360$ °C, and GHSV = 1000 mL $g^{-1}$ $h^{-1}$. The method for analyzing reactants and products was the same as that for the $CO_2$ hydrogenation. The conversion rate of CO and the selectivity of $CO_2$, methane, $C_{2-4}$ HCs, $C_5^+$ HCs, MeOH, DME, and $C_2^+$ Oxys were calculated with the following equations.

$$\text{Conversion rate of CO} = [F_{in}(CO) - F_{out}(CO)]/V_m/m_{GaN} \quad (8)$$

$$\text{Selectivity of } A = F_{out}(A)/[F_{in}(CO) - F_{out}(CO)] \times 100\% \quad (9)$$

where $F_{in}$ is the flow rate in the inlet and $F_{out}$ is the flow rate in the outlet. $A$ is one of the products in the CO hydrogenation reaction. $V_m$ is the molar volume of an ideal gas at the standard temperature and pressure, which 22.4 L/mol is used for calculations. $m_{GaN}$ is the weight of GaN in the catalyst bed.

**Computational details**. All spin-polarized calculations were performed using Vienna Ab-initio Simulation Packages[36,37]. The Perdew–Burke–Ernzerhof generalized gradient approximation functional was used for the exchange-correlation potential[38] and the projected augmented wave potential was applied to describe the ion-electron interaction[39]. The cutoff energy for the plane-wave basis was set to be 400 eV. The $3d$ levels for Ga were explicitly treated, and the DFT + U method with an effective $U$ of 3.9 eV was employed for the $3d$ orbitals[40,41]. The surfaces of GaN (110), (100), and the (110)/(100) interface were sampled using a $2 \times 2 \times 1$ Monkhorst–Pack k-point mesh[42]. The van der Waals dispersion forces were also considered using the zero damping DFT-D3 method of Grimme[43]. All structures were optimized until the forces on each ion were smaller than 0.02 eV $\text{Å}^{-1}$, and the convergence criterion for the energy was $10^{-5}$ eV. The transition states of chemical reactions were located through the dimer minimum-mode method combined with a nudged elastic band method[44,45]. The convergence criterion of transition states is 0.05 eV $\text{Å}^{-1}$. All transition states were identified by the vibration analysis. The details on calculating the rate constant of the reaction steps for the $CO_2$ hydrogenation at 360 °C can be found in our previous study[46].

## Data availability
The data supporting the findings of this study are available within the article and its Supplementary Information files. All other relevant source data are available from the corresponding author upon reasonable request.

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

## Acknowledgements

We thank the financial supports from the National Natural Science Foundation of China (21636006, 21603135, 91945301, and 91645203), the Fundamental Research Funds for the Central Universities (GK201901001 and cxtd2017004), and the Shaanxi Creative Talents Promotion Plan-Technological Innovation Team (2019TD-039). The DFT calculations were performed by using the HPC Platform at Xi'an Jiaotong University. The authors thank Dr. Prof. Jun Bao in National Synchrotron Radiation Laboratory, University of Science and Technology of China, for the kind assistance with X-ray absorption fine structure analyses.

## Author contributions

C.L. performed most of the experiments for preparations and evaluations of catalysts and co-wrote the paper. J.-C. K. conducted $NH_3$-TPD characterizations and co-wrote the paper. Z.-Q.H. performed the DFT calculations and drafted the part of the paper. Y.-H.S. and H.-Q.G. performed the characterizations of XPS and XRD, and a part of the catalytic evaluations. J.S. optimized the techniques for the catalyst preparation. Y.-S.X. and J.-X.H. conducted TEM characterizations and CO hydrogenation experiments. C.-R.C. supervised the DFT calculations and co-wrote the paper. Y.W. analyzed the experimental results and co-wrote the paper. Z.-T.L. analyzed the experimental results. Z.-W.L. made the initial discovery, supervised the experiments, and co-wrote the paper. All of the authors discussed the results and revised the paper.

## Competing interests

The authors declare no competing interests.
