## [Peer Review File · Nature Communications]

Reviewers' Comments:

Reviewer #1:

Remarks to the Author:

The manuscript presents the results obtained during direct CO₂ hydrogenation to DME in presence of differently calcined gallium nitride catalysts (different particle size). The catalysts were characterized according several techniques (XRD, XAFS, XPS, NH₃ adsorption isotherms, Py-IR, operando DRIFT measurements, TEM observations), while catalytic data were obtained in a fixed bed reactor under different conditions of H₂/CO₂ ratio, temperature, space velocity, while maintaining constant the reaction pressure at 2.0 MPa. On the whole, the paper is written in a sufficiently clear manner, with characterization and modelling data certainly of very good quality with important information about properties of the GaN catalyst. The message about the unconventional mechanism of this catalyst is certainly elegant and strong, however, what is not fully convincing is the presentation of the whole work.

The Introduction lacks of several authoritative and fresh references, showing not a complete knowledge in the topic field. Some statements are also to be revised, like at line 49 the claim according to which "metallic Cu catalyzes the hydrogenation of CO₂ to methanol" (there is no consensus about the active phase of this step!!!). Besides, if a direct hydrogenation process is carried out from CO₂ to DME, it is unanimously demonstrated that the activity-selectivity pattern of the catalyst cannot be considered as a mere linear combination of its performance in each of the two steps involved in the process (methanol synthesis from CO₂ and methanol dehydration into DME). From this point of view, the runs performed by co-feeding methanol and H₂O at atmospheric pressure are to be reconsidered.

The presentation of many catalytic results is poor, not being well clear the need for removing CO among the formed products. It is obvious that its removal from the product distribution dramatically increases the selectivity of the other compounds. Nevertheless, a thermodynamic reference is missing in the Figures, considering that the range of temperature is higher than that typically operated (<280 °C). Yet, the differences of the H₂/CO₂ ratio (2 instead of 3) obviously result in dramatic changes which require thermodynamic references. In these sense, it is not sure that the reported STY values are really valuable. Comparison with other catalytic systems tested under similar conditions is needed. Furthermore, up and down of some trends in Figures also require an indication of the experimental error behind, while the choice to shift some crucial results in the Suppl. Info does not help reading the whole MS,

Not even all the conclusions of the work are fully convincing, since many results are obtained at low space velocity, far enough from a full kinetic control suitable to assess a superior behaviour or preferential paths under the adopted conditions.

For these reasons, I'm not sure that the MS meets the requirements for submission in this journal, so suggesting a deep revision and a more appropriate resubmission in another journal.

Reviewer #2:

Remarks to the Author:

This is a very interesting manuscript reporting, for the first time, the use of GaN as a catalyst for the direct hydrogenation of CO₂ to DME. Differently from the traditional one-step DME synthesis from CO₂/H₂ over hybrid metal/acid systems (e.g. Cu-based catalyst + H-zeolite), the GaN catalyst produces DME as a primary product (besides CO). The particle size of GaN is shown to strongly affect the catalytic performance which, based on XRD analysis, is rationalized based on the different crystalline planes that are preferentially exposed to reactants. A plausible mechanism (and involved intermediates) for the preferential formation of DME on specific crystal planes of GaN is proposed based on operando DRIFTS and DFT calculations.

Overall, this is an excellent original work revealing a new catalyst and chemistry for CO₂ valorization and, in my opinion, merits publication in Nature Communications with minor revisions according to the

following comments:

1. Abstract (line 20). The word "selectivity" is duplicated in the sentence: "... with a CO-free selectivity selectivity as high as ...".
2. Lines 59-60. Please, revise the following statement: "Thus, taking into these properties ... to DME account" (probably "Thus, into account these properties ..." is the correct expression).
3. For the best GaN catalyst at optimum reaction conditions the authors report a STY of DME of 2.9 mmol/(g·h). In order to put this productivity value into context, it would be interesting to compare it with that obtained for some of the best-performing hybrid systems (e.g., Cu-based catalyst + zeolite) reported in the literature (such a comparison could be included in the Supporting Information).
4. Lines 101-102. "..., we synthesized bulk GaN with different crystal sizes by calcining the mixture of gallium nitrate and melamine." I would suggest, for the sake of clarity, to specify here that different crystal sizes were produced by changing the duration of the calcination treatment, as one may infer from the experimental section.
5. Line 115. "... (FID), DME is exclusively the main product." Since DME is not the only product detected by the FID, I suggest to remove the word "exclusively" to avoid confusion.
6. In line 123, it is mentioned the existence of an "induction period" of about 12 h before a stable performance is achieved. In Figure 4, however, one can see significant fluctuations in both CO₂ conversion and product selectivities during the first 12 h of reaction. Can the authors discard analytical errors as the origin of this anomalous behavior? It seems questionable to me that this is a true "induction period" since, according to XPS, the nature of Ga and N species was the same in the fresh and spent samples (Fig. S1, Table S1).
7. Lines 159 and 274. "Absorption" should be replaced by "adsorption".
8. Table S3. The sum of product distributions does not always adds up to 100%. Please, check.

Reviewer #3:

Remarks to the Author:

The paper of Liu and co-workers describes a catalytic system for the CO₂ reduction to DME. It is an extensive, multipart study, in which authors use different techniques to shed light on the efficiency and the mechanism of the process.

My main objection is the organization of the manuscript. The present version is very difficult to follow, and in my opinion it is because the authors present the full research paper in the form of a short communication. A significant portion of the results has been moved to the supplementary information, and there are too many references in the manuscript to the supplement. In this case, the manuscript cannot stand on its own - the supporting information is often essential in the discussion. For instance - the manuscript does not even contain even the most basic information on the computational part (such as the functional used). My advice is to either rewrite the manuscript with the better suited form in mind, with the divisions into chapters and sections - as this will greatly improve the clarity of presentation; or focus the description on a strong point and supporting evidence.

Another issue is the analysis of the computational part. While the values of the adsorption energies bring useful information to the topic of study (although the observations are sometimes counterintuitive), the mechanistic study is very much incomplete. The main conclusion seems to be

the competitive pathways via carboxyl and formate lead to the same -CH₃ and -OH intermediates coadsorbed, regardless on the pathway. This does not explain the mechanism in any way, and in my view it only makes the computational part of the study look like it is an unnecessary addition. No other hydrogenation steps have been investigated, no information on the different modes of adsorption of the intermediates has been provided, no analysis of the different character of adsorbed hydrogens has been carried out.

Contrary to that, the experimental part seems convincing and carried out with care. Sometimes it makes the impression of being too extensive - for instance omission of the paragraph devoted to carbonate promoters would not lead to the lesser scientific value of the research, and in my view it would only increase the clarity of presentation mentioned above.

Overall, the paper seems to be an example of those that carry too much information instead of sending one clear message. The manuscript has a potential, and the description of the reactivity of the GaN system is a valuable contribution to the field, but more work is needed to make it meet the standard.

The point-by-point response to Reviewer 1

Comment: The manuscript presents the results obtained during direct CO₂ hydrogenation to DME in presence of differently calcined gallium nitride catalysts (different particle size). The catalysts were characterized according several techniques (XRD, XAFS, XPS, NH₃ adsorption isotherms, Py-IR, operando DRIFT measurements, TEM observations), while catalytic data were obtained in a fixed bed reactor under different conditions of H₂/CO₂ ratio, temperature, space velocity, while maintaining constant the reaction pressure at 2.0 MPa. On the whole, the paper is written in a sufficiently clear manner, with characterization and modelling data certainly of very good quality with important information about properties of the GaN catalyst. The message about the unconventional mechanism of this catalyst is certainly elegant and strong, however, what is not fully convincing is the presentation of the whole work.

Response: Thank you very much for your kind evaluation and positive comments on our manuscript. We have revised the manuscript based on your and the other reviewers' valuable comments, and the point-by-point reply to each of your specific comment is given as follows.

Comment: The Introduction lacks of several authoritative and fresh references, showing not a complete knowledge in the topic field.

Response: We accept your suggestion by adding the up-to-date references published in 2019–2020, which are highlighted in the revised manuscript and are listed as follows for your convenience.

1. Liu, Y., Deng, D. & Bao, X. Catalysis for Selected C1 Chemistry. *Chem.* 6, 2497–2514 (2020).
2. van Kampen, J., Booneveld, S., Boon, J., Vente, J., van Sint Annaland, M. Experimental validation of pressure swing regeneration for faster cycling in sorption enhanced dimethyl ether synthesis. *Chem. Commun.* 56, 13540–13541 (2020)
3. Carvalho, D., Almeida, G., Monteiro, R., Mota, C. Hydrogenation of CO₂ to Methanol and Dimethyl Ether over a Bifunctional Cu·ZnO Catalyst Impregnated on Modified γ -Alumina. *Energy Fuels* 34, 7269–7274 (2020)
4. Pushpakaran, B. N., Subburaj, A. S. & Bayne, S. B. Commercial GaN-Based Power Electronic Systems: A Review. *J. Electron. Mat.* 49, 6247–6262 (2020).
5. Yamahara, K. et al. Ultrafast spatiotemporal photocarrier dynamics near GaN surfaces studied by terahertz emission spectroscopy. *Sci. Rep.* 10, 14633 (2020).
6. Dutta, K. Shahryari, M. & Kopyscinski, J. Direct Nonoxidative Methane Coupling to Ethylene over Gallium Nitride: A Catalyst Regeneration Study. *Ind. Eng. Chem. Res.* 59, 4245–4256 (2020).
7. Aloise, A. et al. Desilicated ZSM-5 zeolite: Catalytic performances assessment in methanol to DME dehydration. *Micropo. Mesopo. Mat.* 302, 110198 (2020)
8. Catizzone, E. et al. Catalytic application of ferrierite nanocrystals in vapour-phase dehydration of methanol to dimethyl ether. *Appl. Catal. B. Environ.* 243, 273–282 (2019).

Comment: Some statements are also to be revised, like at line 49 the claim according to which “metallic Cu catalyzes the hydrogenation of CO₂ to methanol” (there is no consensus about the active phase of this step!!!).

Response: Thank you and we agree with you. Indeed, the metallic Cu (Cu⁰) is not unambiguously accepted as the only active phase for the CO₂ hydrogenation over the Cu-based catalysts although the Cu⁰ phase has been revealed to be one of the most active sites for the CO₂ hydrogenation as reviewed in the reference (Wang, W., Wang, S., Ma, X. & Gong, J. L. Recent advances in catalytic hydrogenation of carbon dioxide. Chem. Soc. Rev. 40, 3703–3727 (2011)). For accuracy, the sentence is modified as “the metal based catalysts such as Cu/ZnO/Al₂O₃ catalyze the hydrogenation of CO₂ to methanol...” (Lines 48–49 in the revised manuscript with yellow highlights). Moreover, the expression is applied in the cases throughout the manuscript, which is also highlighted in yellow color.

Comment: Besides, if a direct hydrogenation process is carried out from CO₂ to DME, it is unanimously demonstrated that the activity-selectivity pattern of the catalyst cannot be considered as a mere linear combination of its performance in each of the two steps involved in the process (methanol synthesis from CO₂ and methanol dehydration into DME). From this point of view, the runs performed by co-feeding methanol and H₂O at atmospheric pressure are to be reconsidered.

Response: Thank you, and we accept your recommendation, which new experiments by using methanol or methanol + H₂O as the reactants are performed, respectively. Indeed, the two steps involved in the reaction of CO₂-to-DME over the typical hybrid catalysts should not be linearly combined over the GaN catalysts. Therefore, we investigate the reaction pathway of CO₂ hydrogenation over the GaN catalysts through catalytic evaluations under different W/F (Figure 6a) and TPSR (Figure 6b). These results demonstrate that DME is the primary product and methanol is the secondary product, of which a different reaction pathway is proposed based on the DFT calculations.

To directly confirm the source of methanol as a secondary product, the experiment using DME and H₂O as reactants is performed to realize the hydrolysis of DME to methanol ($\text{CH}_3\text{OCH}_3 + \text{H}_2\text{O} = 2\text{CH}_3\text{OH}$), and the results are given in Figure R1a as follows and Figure S8a in the revised SI. At a reaction temperature of 320 °C, methanol is the only product, indicating that the hydrolysis of DME occurs exclusively over the GaN catalyst. However, when the reaction is performed at a high temperature of 360 °C, a comparable selectivity of methanol and CO₂ is obtained together with the production of H₂ (Not show), indicating that both the hydrolysis of DME and the steam reforming of DME (SRD, $\text{CH}_3\text{OCH}_3 + 3\text{H}_2\text{O} = 2\text{CO}_2 + 6\text{H}_2$) occur in a comparable extent.

To unambiguously confirm any other possibilities by combining your recommendation and our consideration, additional experiments by using methanol or methanol + H₂O as the reactants are performed, respectively, and the results are given in Figures S8b-c in the revised SI, and are appended as follows in Figures R1b-c. In the case of only methanol as the reactant (Figure R1b), irrespective of the reaction temperatures, DME from the dehydration of methanol ($2\text{CH}_3\text{OH} = \text{CH}_3\text{OCH}_3 + \text{H}_2\text{O}$) is always the main

product. Moreover, a higher reaction temperature increases the selectivity of both CO and CO₂. Thus, the steam reforming of methanol (SRM, CH₃OH + H₂O = CO₂ + 3H₂) and the decomposition of methanol (CH₃OH = CO + 2H₂) occurs significantly at a higher reaction temperature, which is confirmed from the simultaneous release of H₂. On the contrary, when methanol and H₂O are co-fed as the reactants (Figure R1c), the very low selectivity of DME of 0.20 and 0.16% is obtained at the reaction temperatures of 320 and 360 °C, respectively. Moreover, CO and CO₂ are the main products irrespective of the reaction temperatures, and the selectivity of CO is significantly increased with increasing the reaction temperature. In fact, supported Cu is a good catalyst for SRM and the hybrid of supported Cu and a solid acid is also a good catalyst for the SRD reaction, the extent of which is strongly dependent on the specific catalyst and the reaction temperature. For more details on the SRD catalytic network, please see our published works, e.g., Chemical Engineering Journal 187 (2012) 299-305 and Catalysis Today 351 (2020) 68-74. These contents with the detailed experimental procedure, results and discussion are added in Lines 187–195 in the revised manuscript and Section 7 in SI.

Figure R1. The conversion and the selectivity of different products for the experiments with the reactants of DME + H₂O (a), only MeOH (b), and MeOH + H₂O (c) over the GaN-26.6 catalyst under the conditions of $T = 320$ or 360 °C, $P = 0.1$ MPa, and $GHSV = 4000$ mL·g⁻¹·h⁻¹.

Comment: The presentation of many catalytic results is poor, not being well clear the need for removing CO among the formed products. It is obvious that its removal from the product distribution dramatically increases the selectivity of the other compounds.

Response: Sorry for this. In fact, CO is produced via the reverse water-gas-shift (RWGS) reaction. Moreover, the CO₂ hydrogenation and the RWGS are parallel reactions over the GaN catalysts as revealed from much low activity for the CO hydrogenation (Figure 5). This is also supported from the TPSR results given in Figure 6b, i.e., the further conversion of CO hardly occurs (corresponding discussions please see in lines 213–216). Therefore, the selectivity of products involving CO can indicate the catalytic performance via the different pathways, i.e. the direct CO₂ hydrogenation or the RWGS reaction. However, the RWGS reaction does not significantly affect the distributions among the hydrogenated products, i.e. methane, DME, methanol, etc. Thus, to calculate the distributions in hydrogenated products, the RWGS reaction should be excluded. Based on these considerations, the selectivity of different products is expressed including CO while the product distribution is presented without CO. For clarity and conciseness, the word “CO-free” is added in the cases without considering CO in the products in the revised manuscript,

which is highlighted in yellow color.

Comment: Nevertheless, a thermodynamic reference is missing in the Figures, considering that the range of temperature is higher than that typically operated (<280 °C). Yet, the differences of the H₂/CO₂ ratio (2 instead of 3) obviously result in dramatic changes which require thermodynamic references. In these sense, it is not sure that the reported STY values are really valuable. Comparison with other catalytic systems tested under similar conditions is needed.

Response: Thank you. Indeed, the optimized reaction temperature for the typical Cu-based hybrid catalysts, is 240–280 °C, which is the balance of the thermodynamics limitation and the sintering of Cu at higher temperatures. Thermodynamically, as reported in the references, e.g., Chem. Eng. Technol. 37 (2014) 1765–1777, there exists a minimum CO₂ conversion with increasing the temperature, i.e., first decrease followed by the increase of the CO₂ conversion, the extent of which is clearly dependent on the H₂/CO₂ ratio and the pressure as given in Figure R2. Moreover, our calculations by using Aspen plus (Figure R3) are consistent with the reported results. Under the conditions of 2.0 MPa and 360 °C, the equilibrium CO₂ conversion is 31.1% and 26.6% with a H₂/CO₂ molar ratio of 3 and 2, respectively (Referring Figure R2 and Figures R3b–c). For the reaction results over GaN catalysts as given in Figure 3 in the manuscript, the maximum CO₂ conversion under the reaction conditions of 2.0 MPa and 360 °C is 16.8% and 10.6% with a H₂/CO₂ molar ratio of 3 and 2, respectively, which is sufficiently lower than that of the equilibrium CO₂ conversion. That is why there are still some reports on the CO₂ hydrogenation to DME over the typical Cu-based catalysts operated at the reaction temperatures of higher than 280 °C (Please see Table R1). In these cases, the increasing temperature leads to the significant decrease in the selectivity of DME and clear increase in the selectivity of hydrocarbons over the typical Cu-based catalysts. Based on your comments, we gave the references of catalytic evaluations over Cu-based catalysts at low and high temperatures in Table S4, and made the comparison between Cu-based and GaN catalysts under similar reaction conditions as given in the following Table R1. It is observable that at the temperature of higher than or equal to 300 °C, the pressure of 2–5 MPa, GHSV of lower than 3600 mL·g⁻¹·h⁻¹ and H₂/CO₂ = 3, the GaN catalyst exhibits a lower CO₂ conversion but a much higher selectivity of DME than the Cu-based hybrid catalysts. Moreover, the STY of DME over GaN catalysts at 360 °C is similar as those over Cu-based hybrid catalysts at 300–350 °C. Noteworthy, the CaCO₃-GaN catalyst exhibits a much higher STY of DME than the Cu-based catalysts at higher reaction temperature, lower pressure and similar GHSV. Therefore, the GaN catalysts favors the selective hydrogenation of CO₂ to DME at higher reaction temperature, and the promotion of CaCO₃ can further enhance the catalytic performance of CO₂-to-DME.

Figure R2. The thermodynamics results adapted from Chem. Eng. Technol. 37 (2014) 1765–1777. (a) Effect of H_2/CO_2 molar ratios and temperatures on the CO_2 conversion at a fixed pressure of 3 MPa; (b) Effect of pressures and temperatures the CO_2 conversion at a fixed H_2/CO_2 molar ratio = 3.

Figure R3. The thermodynamics results calculated with the software of Aspen plus for the reaction of $CO_2 + H_2 = DME$ (a) and $CO_2 + H_2 = DME$ plus $CO_2 + H_2 = CO + H_2O$ at a H_2/CO_2 molar ratio of 3 (b) and 2 (c) (The pressure is kept at 2.0 MPa).

Table R1. Comparison of Cu-based hybrid catalysts and GaN catalysts in hydrogenation of CO_2 to DME under the similar reaction

Catalyst	Reaction conditions				Catalytic performance ^[a]			Reference
	T	P	GHSV	H_2/CO_2	CO_2	DME	STY_{DME}	
	(°C)	(MPa)	($mL \cdot g^{-1} \cdot h^{-1}$)		Conv. (%)	Sele. (%)	($mmol \cdot g^{-1} \cdot h^{-1}$)	
CuZnAlO _x /HZSM-5	300	3.0	1800	3	29.5	24.1	0.92	Appl. Surf. Sci. 345 (2015) 1
CuZrO _x /Montmorillonite	300	4.0	3600	3	14.1	58.5	0.54	React. Kinet. Mech. Catal. 130 (2020) 179
CuZnZrO _x /Y-Zeolite	350	5.0	3000	3	34.1	2.5	0.51	Appl. Catal. A. Gen. 121 (1995) 113
GaN	360	2.0	3000	3	7.5	75.4	0.59	This work
GaN	360	2.0	3000	2	6.3	79.1	0.85	This work
CaCO ₃ -CaN	360	2.0	3000	2	10.7	47.9	2.9	This work

conditions.

^[a] CO_2 Conv.: Conversion of CO_2 ; DME Sele.: selectivity of DME without CO; STY_{DME} : space-time yield of DME.

Comment: Furthermore, up and down of some trends in Figures also require an indication of the experimental error behind, while the choice to shift some crucial results in the Suppl. Info does not help reading the whole MS.

Response: Thank you, and we accept your suggestion by considering the experimental errors. Therefore, error bars are added in Figures 1, 3–6 in the revised manuscript. With the very reasonable experimental errors, the trends are still valid.

Comment: Not even all the conclusions of the work are fully convincing, since many results are obtained at low space velocity, far enough from a full kinetic control suitable to assess a superior behaviour or preferential paths under the adopted conditions.

Response: Thank you, but we cannot fully agree with you. From the previous publications shown in Table R2, the CO₂ hydrogenation to DME over the typical hybrid catalysts are performed with a GHSV of about 1500 and 10000 mL·g⁻¹·h⁻¹. However, increasing GHSV from 4333 to 34666.67 mL·g⁻¹·h⁻¹ decreases the CO₂ conversion from 21.5 to 8.2% and STY of DME from 5.9 to 2.3 mmol·g⁻¹·h⁻¹ (Chem. Eng. J. 348, 713–722 (2018)). Therefore, we employed a medium GHSV of 3000 mL·g⁻¹·h⁻¹ to keep a reasonable CO₂ conversion, and comparable results are obtained (Table S4).

Thermodynamically, as reported in the references, e.g., Chem. Eng. Technol. 37 (2014) 1765–1777, there exists a minimum CO₂ conversion with increasing the temperature, i.e., first decrease followed by the increase of the CO₂ conversion, the extent of which is clearly dependent on the H₂/CO₂ ratio and the pressure as given in Figure R2. Moreover, our calculations by using Aspen plus (Figure R3) are consistent with the reported results. Under the conditions of 2.0 MPa and 360 °C, the equilibrium CO₂ conversion is 31.1% and 26.6% with a H₂/CO₂ molar ratio of 3 and 2, respectively (Referring Figure R2 and Figures R3b–c). For the reaction results over GaN catalysts as given in Figure 3 in the manuscript, the maximum CO₂ conversion under the reaction conditions of 2.0 MPa and 360 °C is 16.8% and 10.6% with a H₂/CO₂ molar ratio of 3 and 2, respectively, which are only about half of the equilibrium CO₂ conversions. Thus, the results in our cases are under kinetically controlled rather than thermodynamically limited.

Considering your concern, the different mechanism, and the new-type of catalyst, the kinetics of the GaN catalyzed CO₂-to-DME is worthy to be studied in our future work.

Table R2. Catalytic results of the hydrogenation of CO₂ to DME over the Cu-based hybrid catalysts under the different GHSV.

Catalyst	Reaction conditions				Catalytic performance ^[a]			Reference
	T (°C)	P (MPa)	GHSV (mL·g ⁻¹ ·h ⁻¹)	H ₂ /CO ₂	CO ₂ Conv. (%)	DME Sele. (%)	STY _{DME} (mmol·g ⁻¹ ·h ⁻¹)	
CuZnZrO _x /HZSM-5	240	3.0	10000	3	13.4	80.8	4.9	Fuel 241 (2019) 695
CuZnZrO _x /Ferrierite	260	5.0	8800	3	26.0	81.3	7.2	J. CO ₂ Util. 18 (2017) 353
CuZnZrO _x /WO ₃ -ZrO ₂	260	3.0	4333	3	21.5	90.0	5.9	Chem. Eng. J. 348 (2018) 713
CuZrO _x /Montmorillonite	300	4.0	3600	3	14.1	58.5	0.54	React. Kinet. Mech. Catal. 130 (2020) 179
CuZnAlO _x /HZSM-5	260	3.0	1500	3	19.2	91.6	1.7	RSC Adv. 8 (2018)30387

^[a] CO₂ Conv.: Conversion of CO₂; DME Sele.: selectivity of DME without CO; STY_{DME}: space-time yield of DME.

Comment: For these reasons, I'm not sure that the MS meets the requirements for submission in this journal, so suggesting a deep revision and a more appropriate resubmission in another journal.

Response: Thank you very much for your patience in reviewing our manuscript. We have modified the manuscript based on your and the other reviewers' comments. With the new experimental and DFT results, as you may see, both the clarity and the quality of the revised manuscript is significantly improved.

The point-by-point response to Reviewer 2

Comment: This is a very interesting manuscript reporting, for the first time, the use of GaN as a catalyst for the direct hydrogenation of CO₂ to DME. Differently from the traditional one-step DME synthesis from CO₂/H₂ over hybrid metal/acid systems (e.g. Cu-based catalyst + H-zeolite), the GaN catalyst produces DME as a primary product (besides CO). The particle size of GaN is shown to strongly affect the catalytic performance which, based on XRD analysis, is rationalized based on the different crystalline planes that are preferentially exposed to reactants. A plausible mechanism (and involved intermediates) for the preferential formation of DME on specific crystal planes of GaN is proposed based on operando DRIFTS and DFT calculations. Overall, this is an excellent original work revealing a new catalyst and chemistry for CO₂ valorization and, in my opinion, merits publication in Nature Communications with minor revisions according to the following comments:

Response: Thank you very much for your positive comments on our manuscript. We have revised the manuscript based on your and the other reviewers' valuable comments, and you may find all the changes in the List of Changes. The point-by-point reply to each of your specific comment is given as follows.

Comment: 1. Abstract (line 20). The word “selectivity” is duplicated in the sentence: “... with a CO-free selectivity selectivity as high as ...”.

Response: Thank you. The sentence has been modified as “...with a CO-free selectivity of as high as about 80%...”in line 20.

Comment: 2. Lines 59-60. Please, revise the following statement: “Thus, taking into these properties ... to DME account” (probably “Thus, into account these properties ...” is the correct expression).

Response: Thank you. the sentence is modified as “considering these properties and the mechanistic understanding of CO₂-to-DME conversion” in lines 59–60.

Comment: 3. For the best GaN catalyst at optimum reaction conditions the authors report a STY of DME of 2.9 mmol/(g·h). In order to put this productivity value into context, it would be interesting to compare it with that obtained for some of the best-performing hybrid systems (e.g., Cu-based catalyst + zeolite) reported in the literature (such a comparison could be included in the Supporting Information).

Response: Thank you, and we agree with you. We added the information about the comparison between Cu and GaN based catalysts in Table S4. In comparison with the typical Cu-based catalysts, under the reaction temperatures of 240–260 °C, the hybrid catalysts exhibit high selectivity of DME and high STY_{DME}. However, with the increasing temperature, the selectivity and STY of DME over the Cu-based hybrid catalysts clearly decrease. At 360 °C, the selectivity of DME over the GaN catalyst is significantly higher than the Cu-based hybrid catalysts, while STY_{DME} is similar as the hybrid catalysts. Furthermore, the CaCO₃-GaN catalyst exhibits even higher STY_{DME} of 2.9 mmol/(g·h) than the Cu-based catalyst (0.51 mmol/(g·h)). Thus, the CO₂-to-DME at higher temperature is more favorable over the GaN catalysts than

the Cu-based catalysts.

Comment: 4. Lines 101-102. "..., we synthesized bulk GaN with different crystal sizes by calcining the mixture of gallium nitrate and melamine." I would suggest, for the sake of clarity, to specify here that different crystal sizes were produced by changing the duration of the calcination treatment, as one may infer from the experimental section.

Response: Thank you. We agree with you. The information about the duration of the calcination treatment has been added. You can find it in line 104, "...by calcining the mixture of gallium nitrate and melamine with the durations of 1–4 hours at 800 °C."

Comment: 5. Line 115. "... (FID), DME is exclusively the main product." Since DME is not the only product detected by the FID, I suggest to remove the word "exclusively" to avoid confusion.

Response: Thank you. We agree with you. The sentence has been modified as "... (FID), DME is the main product, and..." in line 117.

Comment: 6. In line 123, it is mentioned the existence of an "induction period" of about 12 h before a stable performance is achieved. In Figure 4, however, one can see significant fluctuations in both CO₂ conversion and product selectivities during the first 12 h of reaction. Can the authors discard analytical errors as the origin of this anomalous behavior? It seems questionable to me that this is a true "induction period" since, according to XPS, the nature of Ga and N species was the same in the fresh and spent samples (Fig. S1, Table S1).

Response: Thank you. To discard the experimental errors, we repeated the catalytic evaluations and modified the Fig. 4. Now the new catalytic results with the error bars are present. It is observable that the changes of CO₂ conversion and product selectivities are significant during the first 12 h of reaction. Thus, the induction period does exist in the catalytic evaluation. In the case of XPS results, the binding energies for Ga and N are similar before and after the catalytic evaluations (Figure S1), however, we cannot be sure that the surface propriety of GaN catalyst is unchangeable during the catalytic evaluations. Furthermore, the induction period can be explained by the results of NH₃-TPD over the fresh and spent catalysts (Figure 6a and Table S3). You can also find the results in Figure R4. The results indicate that a gradual loss of the acidity with increasing the time on stream, and is consistent with the decreased CO₂ conversion during the initial induction period. Moreover, the secondary reactions of DME and/or methanol to HCs may be inhibited due to the loss of the acidity, leading to the increased DME selectivity and the decreased selectivity of HCs during the induction period (Figure 4).

Figure R4. NH₃-TPD profiles of the fresh and spent GaN-26.6 catalysts. The spent GaN-26.6 catalyst was obtained under the conditions of P = 2.0 MPa, T = 360 °C, H₂/CO₂ = 2, GHSV = 3000 mL·g⁻¹·h⁻¹ and time of stream = 100 h.

Comment: 7. Lines 159 and 274. “Absorption” should be replaced by “adsorption”.

Response: Thank you. We agree with you. The word has been modified. For the first place, the corresponding sentence was deleted based on the comments of another reviewer. For the other place, the sentence was modified as “...the slight difference between the adsorption energy of H₂ and...” and was shown in the line 273.

Comment: 8. Table S3. The sum of product distributions does not always add up to 100%. Please, check.

Response: Thank you. Table S3 used to represent the catalytic results over GaN promoted by different promoters. In fact, the carbon balances for all the catalytic evaluations in this work are always better than 95%, indicating that the results of product selectivities are convincing. However, based on the comments of another reviewer, the catalytic results over promoted GaN are removed.

The point-by-point response to Reviewer 3

Comment: The paper of Liu and co-workers describes a catalytic system for the CO₂ reduction to DME. It is an extensive, multipart study, in which authors use different techniques to shed light on the efficiency and the mechanism of the process.

Response: Thank you very much for your kind evaluation and positive comments on our manuscript. We have revised the manuscript based on your and the other reviewers' valuable comments, and the point-by-point reply to each of your specific comment is given as follows.

Comment: My main objection is the organization of the manuscript. The present version is very difficult to follow, and in my opinion it is because the authors present the full research paper in the form of a short communication. A significant portion of the results has been moved to the supplementary information, and there are too many references in the manuscript to the supplement. In this case, the manuscript cannot stand on its own - the supporting information is often essential in the discussion. For instance - the manuscript does not even contain even the most basic information on the computational part (such as the functional used). My advice is to either rewrite the manuscript with the better suited form in mind, with the divisions into chapters and sections - as this will greatly improve the clarity of presentation; or focus the description on a strong point and supporting evidence.

Response: Thank you, and we are sorry about this. We accept your suggestion, and the modifications on the clarity of the manuscript structure and the independence and integrity of the manuscript are made as follows: 1) To avoid citing too many figures from SI, the Figures are re-organized with clarity and integrity as shown in Figure 3c-h and Figure 5 in the revised manuscript; 2) The computational details are added in the section of Methods with a subsection of Computational details in the revised manuscript; 3) The title of the section or the subject phrase to the paragraph are added throughout the manuscript. All of these changes are highlighted with yellow color in the revised manuscript. With these changes, as you may see, the clarity and the integrity of the manuscript is significantly improved.

Comment: Another issue is the analysis of the computational part. While the values of the adsorption energies bring useful information to the topic of study (although the observations are sometimes counterintuitive), the mechanistic study is very much incomplete. The main conclusion seems to be the competitive pathways via carboxyl and formate lead to the same -CH₃ and -OH intermediates co-adsorbed, regardless on the pathway. This does not explain the mechanism in any way, and in my view it only makes the computational part of the study look like it is an unnecessary addition. No other hydrogenation steps have been investigated, no information on the different modes of adsorption of the intermediates has been provides, no analysis of the different character of adsorbed hydrogens has been carried out.

Response: Thank you very much for the valuable comments. Based on your comments, the DFT calculations have been greatly improved. The adsorption of different intermediates during the CO₂ hydrogenation is systematically studied and the results are listed in Figure S16. In addition, the reaction

energies and activation energies for various elementary steps are calculated. The whole pathway of CO₂ hydrogenation to DME is shown in Figure 7 (Figure R5 below), Figures S13–S21 and Tables S7–S8. In particular, we find that over the (110) plane the activated CO₂ is preferable to be hydrogenated to the formate species while over the (100) plane CO₂ is preferable to be hydrogenated to the carboxyl species, due to the different activation energies (Figure S17). Subsequently, over the (100) plane, the formed carboxyl species are hydrogenated and dehydrated, and finally converted to the methyl species. The DFT results also indicate that DME is formed via the coupling of the formate and the methyl over the (100)/(110) interface. More importantly, during the whole pathway, the methoxyl (CH₃O*) species are absent, which is consistent with the results of DRIFTS (Table S6). This indicates methanol can hardly be formed from the hydrogenation of CH₃O* over GaN. Even though the CH₂OH* species are present, the hydrogenation of CH₂OH* to methanol is difficult due to a high activation energy (1.28 eV). Instead, the CH₂OH* is preferable to be dissociated into CH₂* and OH* with a much lower activation energy of 0.26 eV, which provides the precursor for the formation of CH₃*. Therefore, the improved DFT calculation results not only provide a reasonable reaction mechanism for the formation of DME but also explain well why DME rather than methanol is formed as the primary product on GaN. The theoretical and experimental results together contribute to the comprehensive understanding of the CO₂-to-DME on GaN catalysts.

Figure R5. The DFT calculation results. (a) Gibbs free energy diagram of the CO₂ hydrogenation to methyl (CH₃*) on the GaN(100) surface. (b) Gibbs free energy diagram of the CO₂ hydrogenation to formate (HCOO*) on the GaN(110) surface. (c) Gibbs free energy diagram for the coupling of HCOO* and CH₃* to DME on the (110)/(100) interface.

Comment: Contrary to that, the experimental part seems convincing and carried out with care. Sometimes it makes the impression of being too extensive - for instance omission of the paragraph devoted to carbonate promoters would not lead to the lesser scientific value of the research, and in my view it would only increase the clarity of presentation mentioned above.

Response: Thank you. We accept your suggestion, and the contents devoted to the carbonate promoters are greatly simplified by keeping the necessary results over the CaCO_3 -GaN catalyst. The modified sentences are highlighted in Lines 158–162 of the revised manuscript with yellow highlights.

Comment: Overall, the paper seems to be an example of those that carry too much information instead of sending one clear message. The manuscript has a potential, and the description of the reactivity of the GaN system is a valuable contribution to the field, but more work is needed to make it meet the standard.

Response: Thank you for your positive comments and valuable suggestions. Together with the other reviewers' comments, the additional experiments and detailed DFT calculations are performed, and the results are added in the revised manuscript with yellow highlights. With the additional data and the modifications, the quality and clarity of the revised manuscript is significantly improved.

Reviewers' Comments:

Reviewer #1:

Remarks to the Author:

After revision, the quality of MS appears to be significantly improved, being most of the points arisen convincingly addressed.

However, what is not fully convincing yet is the significance of the values obtained (i.e., STY(DME) values on GaN incredibly low – more than one order of magnitude - if compared with literature data obtained at lower temperature on copper-based catalysts!!!) as well as the presentation of many results (CO is the main product in the adopted conditions and every attempt to show something of different is misleading), comments about thermodynamics (it is confirmed that, in the range of temperature chosen, DME formation is hindered), conditions selected for experiments (i.e., range of temperature).

For these reasons, I cannot express a positive evaluation for this MS, definitely suggesting its rejection.

Reviewer #2:

Remarks to the Author:

The authors have provided suitable answers to my (minor) comments and made the required changes in the manuscript. Also, in my view, the authors have reasonably addressed the main issues raised by the other reviewers and, in consequence, have significantly improved the overall quality and clarity of the manuscript. Therefore, I recommend its acceptance for publication.

I have only a few additional minor comments on formal/grammar aspects that could probably be amended at the editing stage if finally accepted:

- Some figures are placed before they are mentioned in the text.
- In line 75, it should be "a much lower GHSV" instead of "a much low GHSV".
- Lines 296-297: "is significantly more favorable than".
- Lines 341-342: "HCs and oxygenates including methanol are produced"

Reviewer #3:

Remarks to the Author:

The remarks made with respect to the original submission have been addressed by the authors. The computational part has been significantly improved, and in my view it provides sufficient support for the experimental part. I am pleased to recommend the revised version of the manuscript for publication.

The point-by-point response to Reviewer #1

Comment: After revision, the quality of MS appears to be significantly improved, being most of the points arisen convincingly addressed.

Response: Thank you very much for your kind evaluation and positive comments on our manuscript, and the point-by-point reply to each of your specific comments is given as follows.

Comment: However, what is not fully convincing yet is the significance of the values obtained (i.e., STY(DME) values on GaN incredibly low – more than one order of magnitude - if compared with literature data obtained at lower temperature on copper-based catalysts!!!) as well as the presentation of many results (CO is the main product in the adopted conditions and every attempt to show something of different is misleading), comments about thermodynamics (it is confirmed that, in the range of temperature chosen, DME formation is hindered), conditions selected for experiments (i.e., range of temperature). For these reasons, I cannot express a positive evaluation for this MS, definitely suggesting its rejection.

Response: Thank you so much for your kind comments on our work. However, we cannot fully agree with you.

As stated clearly in the manuscript, the importance of our work lies in (1) Wurtzite-structure GaN is active, stable, and selective for catalyzing the direct hydrogenation of CO₂ to DME as a primary product, which is different from the traditional route of the DME synthesis over a hybrid catalyst via the methanol intermediate; (2) The CO-free selectivity of DME is as high as around 80%, and the highest space-time yield of DME is 2.9 mmol·g_{GaN}⁻¹·h⁻¹; (3) A reasonable mechanism via the coupling of CH₃^{*} and HCOO^{*} is proposed based on the steady-state and transient experimental results, operando DRIFTS, and DFT calculations; (4) The activity of GaN for the hydrogenation of CO₂ is much higher than that for the hydrogenation of CO although the product distributions are very similar. Thus, these findings may open up a new catalytic route for the directly efficient utilization of CO₂.

Regarding your comment "i.e., STY(DME) values on GaN incredibly low – more than one order of magnitude - if compared with literature data obtained at lower temperature on copper-based catalysts!!!", it is not the case if the typically reported and our results are examined as show in Table S4. For your convenience, Table S4 is appended as follows.

Table S4. List for the hydrogenation of CO₂ to DME over the reported Cu-based hybrid catalysts and GaN catalysts

Catalyst	Reaction conditions				Catalytic performance			Reference
	T (°C)	P (MPa)	GHSV (mL·g ⁻¹ ·h ⁻¹)	H ₂ /CO ₂	CO ₂ Conversion (%)	DME Selectivity (%)	STY _{DME} (mmol·g ⁻¹ ·h ⁻¹)	
CuZnZrO _x /Ferrierite	260	5.0	8800	3	26.0	81.3	7.2	[9]
CuZnZrO _x /HZSM-5	240	3.0	10000	3	13.4	80.8	4.9	[10]
CuZnZrO _x /WO ₃ -ZrO ₂	260	3.0	4333	3	21.5	90.0	5.9	[11]
CuZnAlO _x /HZSM-5	240	2.8	1525	3	21.4	86.0	6.2	[12]
CuZnAlO _x /HZSM-5	260	3.0	1500	3	19.2	91.6	1.7	[13]
CuZnAlO _x /HZSM-5	300	3.0	1800	3	29.5	24.1	0.92	[14]
CuZrO _x /Montmorillonite	300	4.0	3600	3	14.1	58.5	0.54	[15]
CuZnZrO _x /Y-Zeolite	350	5.0	3000	3	34.1	2.5	0.51	[16]
GaN	360	2.0	3000	3	7.5	75.4	0.56	This work
GaN	360	2.0	3000	2	6.3	79.1	0.85	This work
CaCO ₃ -CaN	360	2.0	3000	2	10.7	47.9	2.9	This work

Without considering the varied reaction parameters, as shown in Table S4, the space-time yield of

DME over GaN catalysts is $0.56 \sim 2.9 \text{ mmol}\cdot\text{g}^{-1}\cdot\text{h}^{-1}$, which is very comparable with those over the Cu-based hybrid catalysts, i.e., $0.51 \sim 7.2 \text{ mmol}\cdot\text{g}^{-1}\cdot\text{h}^{-1}$. If the highest space-time yield of DME over the Cu-based hybrid catalyst ($7.2 \text{ mmol}\cdot\text{g}^{-1}\cdot\text{h}^{-1}$), is concerned, it is achieved at a favorably higher pressure of 5.0 MPa, which is still less than three times higher than that over the GaN catalyst at an unfavorably lower pressure of 2.0 MPa ($2.9 \text{ mmol}\cdot\text{g}^{-1}\cdot\text{h}^{-1}$). More importantly, as a new catalyst, there is still a large room for the further optimization of GaN, which is worthy to be done in the near future.

In the case of your comment of "the presentation of many results (CO is the main product in the adopted conditions and every attempt to show something of different is misleading), comments about thermodynamics (it is confirmed that, in the range of temperature chosen, DME formation is hindered), conditions selected for experiments (i.e., range of temperature)", the most important finding of our work is that GaN is an active, stable, and selective catalyst for the direct hydrogenation of CO_2 to DME as a primary product and the reaction proceeds via the coupling of CH_3^* and HCOO^* , which is completely different from the traditional route of the DME synthesis over a hybrid catalyst via the methanol intermediate. Definitely, the further optimization of the GaN catalyst and the reaction conditions are very necessary, and improved catalytic performance for CO_2 -to-DME is reasonably expected.

Indeed, we agree with you that the formation of CO is thermodynamically favored at higher temperatures while the formation of DME is thermodynamically favored at lower temperatures. Thus, the improvement of the lower-temperature performance of the GaN catalyst is unquestionably one of the most important work for studies in the near future. Fortunately, a reasonable mechanism is proposed in our work, which can provide valuable guidelines for the catalyst optimization.

As a common practice in the references, the CO-free selectivity of DME over Cu-based hybrid catalysts is reported for better understanding the catalytic process. Indeed, as clearly stated in the manuscript, we do not deny the high selectivity of CO over the GaN catalyst at a higher reaction temperature of 360°C ($\sim 60\%$). However, it is still very reasonable in comparison with those over the Cu-based hybrid catalyst reported in the references, e.g., the selectivity of CO = 56.1% at 240°C (Appl. Catal. B. 162, 57–65 (2015)); the selectivity of CO = 69.8% at 260°C (Chem. Eng. J. 348, 713–722 (2018)); the selectivity of CO = 79.5% at 300°C (React. Kinet. Mech. Catal. 130, 179–194 (2020)). More importantly, even at a higher reaction temperature of 360°C , the highest space-time yield of DME of $2.9 \text{ mmol}\cdot\text{g}^{-1}\cdot\text{h}^{-1}$ is achieved over the GaN catalyst, which is clearly higher than some of those over the Cu-based hybrid catalysts, e.g., the space-time yields of DME of $1.7 \text{ mmol}\cdot\text{g}^{-1}\cdot\text{h}^{-1}$ at 260°C and $0.51 \text{ mmol}\cdot\text{g}^{-1}\cdot\text{h}^{-1}$ at 350°C as given in Table S4. Thus, GaN is not only a new but also an efficient catalyst for the direct hydrogenation of CO_2 to DME.

The point-by-point response to Reviewer #2

Comment: The authors have provided suitable answers to my (minor) comments and made the required changes in the manuscript. Also, in my view, the authors have reasonably addressed the main issues raised by the other reviewers and, in consequence, have significantly improved the overall quality and clarity of the manuscript. Therefore, I recommend its acceptance for publication.

Response: Thank you very much for your recommended acceptance of our work for publication. Your kind evaluation and positive comments on our manuscript are highly appreciated, and the point-by-point reply to each of your specific comments is given as follows.

Comment: I have only a few additional minor comments on formal/grammar aspects that could probably be amended at the editing stage if finally accepted:

Response: Thank you very much for your careful reading and corrections of the formal/grammar errors in our manuscript.

Comment: - Some figures are placed before they are mentioned in the text.

Response: Thank you, and we accept your comments by arranging all of the Figures after the text mentioned for the first time, which you may see in the revised manuscript.

Comment: - In line 75, it should be "a much lower GHSV" instead of "a much low GHSV".

Response: Thank you and we are sorry for the grammar error. It is corrected in the revised manuscript, which is highlighted in yellow color.

Comment: - Lines 296-297: "is significantly more favorable than".

Response: Thank you and we are sorry for the grammar error. The missed "more" is added in the revised manuscript, which is highlighted in yellow color.

Comment: - Lines 341-342: "HCs and oxygenates including methanol are produced"

Response: Thank you and we are sorry for the grammar error. The right plural form of "be" is substituted for the wrong singular form, which is highlighted in yellow color in the revised manuscript. Moreover, we have carefully checked the typing, grammar, and the English expressions throughout the manuscript, and the corrections are also highlighted in yellow color in the revised manuscript.

The point-by-point response to Reviewer #3

Comment: The remarks made with respect to the original submission have been addressed by the authors. The computational part has been significantly improved, and in my view it provides sufficient support for the experimental part. I am pleased to recommend the revised version of the manuscript for publication.

Response: Thank you very much for your positive comments on our manuscript and the recommendation for publication.